# Inclusive Assessment and Sustainability Dimensions: Pre-Service Teachers' Ideas and Knowledge

**Beatriz Gallego-Noche, Esther García-González \*** **, Rocío Jiménez-Fontana** and **Pilar Azcárate**

Faculty of Education, University of Cádiz, 11519 Puerto Real, Cádiz, Spain
\* Correspondence: esther.garcia@uca.es

**Abstract:** The eco-social crisis urges the transformation of teacher training programmes. It is imperative for educators to be able to respond to global challenges. This transformation involves working from new perspectives that integrate assessment processes focused on eco-social justice and inclusion from a broad perspective. Bearing this in mind, a training process focused on the theoretical construction of assessment practice from the perspective of Education for Sustainability and Inclusive Education was designed. It is framed within the context of the Master's Degree in Educational Research for Teachers' Professional Development. The study presented, which is of a qualitative and interpretative nature, focuses on knowing the impact of the process in the progression of the participating students' ideas and knowledge construction. The students' group work, two questionnaires of a different nature and a teaching journal, were used to collect data. Likewise, an ad hoc analysis system was developed that allowed transforming and interpreting the data obtained. The results show progress in the ideas and knowledge of the students in aspects related to the socio-environmental and ethical dimension of the assessment, integrating the principles of Education for Sustainability and Inclusive Education in the design of assessment activities in the classroom. The results reflect that the developed training process has influenced the complexity of the ideas of the students with regard to the content worked on. The proposal could hence serve as a reference for other training processes that have similar characteristics.

**Keywords:** inclusive assessment; assessment from the perspective of sustainability; education for sustainability; inclusive education; teacher training



## 1. Introduction

Education for Sustainability (henceforth EfS) is a proposal that focuses on the principles of performance from the perspective of sustainability to respond to the current eco-social crisis. It seeks to develop skills that empower individuals to reflect on their own actions, taking into account their current and future social, cultural, economic and ecological consequences from a local and global perspective [1]. This perspective determines the role given to assessment associated with sustainability.

Within this framework, and after the United Nations Conference for Sustainable Development (Rio+20), EfS assumes the guidelines of the 2030 Agenda. It places emphasis on initiating collective action to search for systemic and global responses. The 2030 Agenda sets out 17 sustainable development goals (SDGs) made up of 169 targets of an integrated and indivisible nature that cover the economic, social and environmental spheres. This research is closely linked to SDG 4, which aims to ensure inclusive and equitable quality education and promote lifelong learning opportunities for all. Particularly related to our study is target 4.7: ensure that all learners acquire the knowledge and skills needed to promote sustainability, including through EfS and sustainable lifestyles: human rights, gender equality, promotion of a culture of peace and non-violence, global citizenship and appreciation of cultural diversity and of culture's contribution to sustainability.

For us, this goal explains a fundamental relationship between EfS and Inclusive Education (henceforth IE), and represent, from our approach, a relevant theoretical-practical support for the purpose of training teachers capable of responding to the challenges posed by today's society [2,3]. We establish this relationship through conceiving IE as an education based on human rights that condemns situations of discrimination and inequality, and starts from the assumption that all people have the capacity to learn. It is the duty of teachers and public administrations work through and towards it. Diversity is understood as an ontological characteristic of the human being, and homogeneity as a fallacy, which only serves the interests of certain groups, and on which hierarchical, unequal, predatory and oppressive systems are built [2,4,5]. Those systems project their domination of the human to the domain of nature [6,7] and makes us understand education from the perspective of the ethics of sustainability and inclusion.

To respond to these demands, we need citizens and professionals that have critical and reflective thinking skills, and the capacity to act in favour of a more sustainable and fair development of our society. This means educational processes have to be redefined [8]. The implemented didactic intervention seeks to reconsider how we are currently training professionals (and citizens), and how we want to train them in the future. Our condition as university teachers and citizens requires a commitment to transformation, which involves improving the circumstances that make it possible [9].

Therefore, the challenge we face as teachers is to build an educational model that questions the prevailing negative values of global problems. In this regard, sustainability, IE and the complex view both imply, emerge as guiding frameworks that shed light on the ideological and ethical dimension. As professionals responsible for the education of future generations, training future teachers plays a key role in achieving this educational model [10].

When EfS and inclusive values permeate teacher training, university teachers themselves shall be committed to ethics and professional responsibility. This implies the need for a change in attitude, methodology, conception of teaching–learning processes and, logically, assessment.

The educational model we advocate for requires consistency in assessment processes. They cannot be considered as an appendix of experiences and teaching and learning processes, or as an element that qualifies and excludes. Assessment is a reality that should be part of the strategies for eco-social justice and inclusion [11,12], enlightened by its ethical and democratic dimension. It is key in teaching–learning processes since, in most cases, it determines all the other activities associated with the intervention and, of course, with learning [13,14]. The way in which assessment is implemented needs to be in harmony with EfS and IE in order to promote the principles associated, which, in turn, affect the development of students' capacities.

Assessment is not a neutral process in any context [15,16]. What and how to assess, the object and the assessment tools (that is, the structure of the assessment system) are aspects conditioned by its meaning (what for) and purpose (why); in other words, the function of the assessment process. The relationship between the moments of assessment: when to assess, permanently related to how to assess and to who is assessing, also needs to be considered.

As explained above, assessment should not be limited to an accrediting function. It should be understood in its context and provide useful information for the adaptation and adjustment of teaching–learning processes to the needs of students, teachers and the process itself, thus improving the quality of the training activity [15,17]. When assessment is understood as (self) regulation [18] and as an element of learning [19] and empowerment [11,13], it acquires a determining role in the configuration of the ethical dimension of students, future professionals, teachers and citizens. It is, therefore, key to design and implement it with the utmost care.

Hence, in this work, the perspectives of EfS and IE are considered as theoretical–practical axes that provide an ethical framework from which to design assessment systems. This framework, which is configured around planetary ethics and inclusive values (Table 1) provides assessment with a broad meaning that guides the understanding of

reality, the construction of criteria to position ourselves and participation in the eco-social transformation [20]. Table 1 is organised, following [2,4], around three dimensions: (1) structures (micro- and macro-political framework), (2) relationships with other people and (3) the human spirit, referring to those values related to emotions and feelings towards IE and EfS.

**Table 1.** Values from the perspective of EfS [21] and IE [2,4].

| Structures | Relationships | Human Spirit |
|:---:|:---:|:---:|
| Equality<br>Justice<br>Equity<br>Eco-centrism<br>Rights<br>Participation<br>Community<br>Freedom | Diversity<br>Non-violence<br>Confidence<br>Compassion<br>Honesty<br>Courage | Cheerfulness<br>Love<br>Hope/Optimism<br>Beauty<br>Creativity |

The purpose of teacher training in the field of assessment is twofold. On the one hand, it seeks to include these values in the design of assessment activities, and on the other hand, to implement the activities designed in the training context itself so that students learn and foster these values. This leads us to consider the following as key principles: democratisation and participation in assessment [16,22,23], diversification of assessment [24], traditional assessment as a barrier and as exclusion [11], assessment as action research [25,26], assessment to adjust teaching to the particularities of the people involved in the process and to those of the process itself [27], assessment as empowerment [13,28,29], assessment for the development of self-efficacy [30,31] and assessment as community-building [32–34], rather than as a competition for an unsustainable labour and economic market.

Starting from the review of the literature, a series of principles are developed to guide us in the design and implementation of assessment around:

- The creation of spaces for dialogue, communication, participation and consensus of the different people involved in the assessment processes, and as a management model of classrooms, schools and society.
- The participation of students and other community agents in decision-making and in the processes involved in assessment.
- The design and development of assessment tasks that acquire sense and meaning in relation to the environment and the socio-environmental reality.
- The implementation of grounded and shared feedback processes.
- The creation and use of shared assessment criteria and tools and rating systems consistent with the values related to sustainability and inclusion.
- The diversification of assessment tasks and tools.
- The establishment of horizontal relationships where the intrinsic value of each person, their ability and their right to always learn is appreciated.
- The community implementation of action research processes that improve the rationality of assessment and educational practices in a given context of eco-social crisis.
- The analysis of circumstances that generate situations of vulnerability, exclusion and deterioration of the environment.
- The creation of spaces based on trust in the capacities of all people, on the honesty and transparency of assessment processes, and on the courage to disobey what is established by law if it is exclusive (standardised assessment tests).
- The assumption of professional responsibility in converting assessment processes into sustainable and inclusive assessment practices, and in identifying and reducing barriers to learning and participation, as well as planning the necessary resources and support for diversity, adopting a cooperative attitude.

If we consider that every action is linked to a system of ideas and knowledge that determines it, designing and developing educational activities that enable this progress and construction of ideas and knowledge is key. However, we are aware that the assessment system as a shared process configured from the presented perspective is very difficult and requires pre-service teachers' ideas to evolve. Future teachers will also need to build complex theoretical-practical knowledge. Accordingly, it will be necessary to know if implementing these educational activities contributes to the pre-service teachers' progression of ideas and construction of complex knowledge. This information will allow us to design training processes that more adequately face the challenge of integrating EfS and IE in assessment systems, the students' future professional actions and the eco-social transformation we need.

The objective of this research is configured as follows: analyse the progression of pre-service teachers' ideas and construction of knowledge of assessment from the perspective of EfS and IE after participating in a training process designed from this approach. It is carried out following the two dimensions considered key in the development of this approach: (1) the socio-environmental and ethical dimension, and (2) the dimension of classroom assessment design. The general objective is further divided into two specific objectives:

- Analyse the development of pre-service teachers' ideas and construction of knowledge regarding the socio-environmental and ethical dimension that affects assessment systems.
- Analyse the development of pre-service teachers' ideas and construction of knowledge regarding the classroom assessment design from the perspective of EfS and IE.

We assume the complexity of this process, as well as the limitations to know the impact of a training process on the students' ideas.

However, we trust the benefits of the study to have a positive impact in the framework of quality, inclusive and equitable education, capable of promoting sustainability through lifelong learning opportunities. Formative assessment has the potential to support teaching and learning in the classroom [35] to the extent that it conditions what and how students learn [36].

## 2. Methods

The study carried out responds to the need to investigate the design and impact of training processes regarding students' knowledge construction and the progression of their ideas, as well as the possibility of promoting the development of those ideas in a collaborative and inclusive manner [37,38]. From our epistemological and methodological position [26], we do not conceive training processes separated from research processes, but as necessary and intersecting processes. The teachers explained this standpoint at the beginning of the training process, establishing a discussion and negotiation about the desire for student involvement. This debate gave rise to the informed consent of the students to collaborate in the research.

The context in which the research is carried out is the same as the one in which the training process unfolds, using the tools of the training process as sources of information.

From this perspective, we designed a qualitative and interpretive study, consistent with our starting position (evaluation from the perspective of EfS and IE) that allows us to approach the students' ideas and construction of knowledge in a real context.

### 2.1. Research Context and Participants

This research was carried out during academic year 2021–2022 at Universidad de Cádiz in the context of the Master's Degree in Educational Research for Teachers' Professional Development, within the course called Problems associated to Assessment. It was organised in eleven sessions of four hours each during the months of January and February. The key purpose of the course was to delve into the perspective of EfS and IE as theoretical and ethical axes from which to design assessment systems. To this end, two fundamental spheres of study were addressed throughout the process: (1) the socio-environmental and

ethical dimension that affects assessment systems, and (2) the classroom assessment design from the perspective of EfS and IE.

With regard to the participants, a total of 24 students, 5 male and 19 female students, aged 22 to 29, participated in the training process and in the research. They were grouped into five work teams to prepare the cooperative tasks. All of them had completed a degree in Primary Education or Early Childhood Education.

Procedure

The design of the training process, based on a constructivist, socio-affective and critical approach [39–42], was structured in three phases that combined individual and group work. At the same time, it constituted the sources of information for the research conducted:

Initial phase: presentation on assessment, personal and professional experiences in the educational path of each student, establishing dynamics for the group to be cohesive, and to create a climate of collaborative work and negotiation regarding what and how students want to learn and be assessed. In this first phase, an initial evaluation of the students' knowledge of aspects related to assessment from the perspective of EfS and IE was carried out by means of a questionnaire (Figure 1) including the following ten questions:

---

1.- Would you get rid of assessment if you could? Do you think it is necessary? Why?

2.- Comment the following sentence: "Teaching and assessment are not neutral tasks, they are processes that have a strong political-ideological and ethical-moral component" (Moreno 2011)

3.- What does assessment contribute to the community, to schools and to families?

4.- How useful is assessment for the teaching–learning process, for teachers and for students?

5.- What kind of information does assessment provide? What do you want it for? What do you/would you do with that information?

6.- What do you understand by inclusive evaluation practices? Justify your answer. Provide examples.

7.- Who makes the decisions about what has to be assessed and how?

8.- Do you think assessment is an activity that is exclusively for teachers? Why? Why not?

9.- What tool do you consider to be the most significant to assess? Justify your answer.

10.- Does the assessment tool depend on what you want to assess? Can you use any tool at any time during the assessment process? Justify your answer. Give examples.

---

**Figure 1.** Initial questionnaire [43].

This initial questionnaire (Figure 1) was configured as one of the sources of information for the study.

Intermediate phase: introduction and description of the global crisis, students' analysis of the relationship of this crisis with educational and assessment systems, and how they can contribute to the transformation towards eco-social justice. In this phase, the students worked together on developing two assessment tasks:

(1)    Development and creative presentation of a simulated situation in which they, as an educational team, had to build an assessment proposal from the perspective of EfS and IE for an early childhood or primary school. In this task, they had to build a theoretical-practical perspective regarding:

a.    the concept of educational assessment

b.    assessment and its relationship with the current socio-economic and political systems and the school context (they were given the description of a specific school)

c.    the role of assessment in a sustainable and inclusive school: key principles and curriculum organisation

d.    standardised assessments

e.    evaluating the school and the teachers (action research)

This task was presented in groups to the whole class, and peer-assessment was performed using a rating scale the students prepared together based on assessment criteria previously agreed upon. This same scale was used to evaluate the work performed by the teacher.

(2) Classroom assessment design from the perspective of EfS and IE based on an initial design. To develop this design, different activities were carried out:

    a. analysis, in small groups, of an educational intervention proposal

    b. esign, in small groups, of an initial assessment proposal for the educational intervention proposal analysed in point a.

    c. whole-class discussion about different proposals to define assessment processes and their impact on teaching–learning processes

    d. analysis of the initial proposal of each small group together with the teachers to adapt the different elements of the assessment processes from the perspective of EfS and IE

    e. whole-class discussion about what they have worked on in small groups regarding the different elements that make up an assessment system from the perspective of EfS and IE so as to build a joint assessment system.

During this phase, the teaching journal was used as a tool to assess and investigate the process. The aim was to adapt the didactic interactions to the needs of the students, teachers and the teaching–learning process itself.

The two assessment tasks (the simulated situation and the design of the assessment process) and the teaching journal are also sources of information for the study presented here.

Final phase: re-drafting of the initial assessment design in light of the information discussed during the intermediate phase. Data were also collected through self-assessment processes, responding to the same individual initial assessment questionnaire; and to a questionnaire with closed- and open-ended questions to rate the development of the knowledge constructed in general, and in each of the dimensions that make up this assessment perspective (Figure 2). The questionnaire consisted of five closed-ended questions in which the students had to rate different aspects of the knowledge worked on in the course from 1 (not achieved) to 10 (fully achieved) and one open-ended question (Figure 2) to complement the information of the previous questions. A total of 24 students responded.

---

1.- Rate your level of knowledge built in the course on a scale from 1 to 10

2.- Rate how the complexity of your knowledge of assessment has evolved during the course on a scale from 1 to 10

3.- Rate your level of knowledge of the ethical, environmental, political and social dimension of assessment on a scale from 1 to 10

4.- Rate your level of knowledge of the inclusive dimension of assessment (community, attention to diversity, research, socio-environmental justice, participation) on a scale from 1 to 10

5.- Rate your level of knowledge of the dimension of classroom assessment design on a scale from 1 to 10

6.- Rate how the complexity of your knowledge of assessment has evolved in a qualitative manner during the course on a scale from 1 to 10

---

**Figure 2.** Final questionnaire to rate the development of knowledge.

This final questionnaire is also considered a source of information for the research carried out.

*2.2. Data Collection Tools and Analysis Procedure*

In order to know the reality under study in a contextualised manner, the use of techniques and tools for the construction of information from a qualitative perspective was chosen [44]. We opted for the analysis of the participants' group work and pre-and post-questionnaires of their ideas, as well as the questionnaire to rate their knowledge. These questionnaires were developed based on the theoretical approaches that we built around assessment from the perspective of EfS and IE. In other words, they were developed to know the students' ideas and knowledge regarding the main dimensions of this approach, explained in the first section of this paper.

The tools used for data gathering and construction, as we pointed out earlier, were the following:

- Individual initial questionnaire (IQ) of open-ended questions adapted from previous research [45], developed in the initial phase of the training process
- Report of the theoretical-practical assessment proposal from the perspective of EfS and IE for a school (school-based educational project, or SEP), prepared in the intermediate phase
- Classroom assessment design from the perspective of EfS and IE (CAD)
- Final individual questionnaire (FQ) and questionnaire to evaluate the progression of knowledge (QEPK), built ad hoc, developed in the final phase of the process

From the research perspective, these four tools are configured as the main sources of information.

As it is a qualitative and action-research study, the reliability and validity of the tools are not considered from statistical procedures, but the criteria of credibility, confirmability and transferability are ensured through reflective processes within the teaching and research team (teachers-students).

For the analysis of the data produced, two dimensions related to the research objectives, the purposes of the training process itself and the theoretical references constructed were formulated: (1) the progression of ideas and (2) the construction of knowledge regarding the relationship between the values that support EfS and IE; social, environmental, political and cultural problems; and how the assessment systems designed and developed from this perspective contribute to the transformation of eco-social conflicts. The two dimensions are described as follows:

DIMENSION 1 (D1). Socio-environmental and ethical dimension: understood as the awareness of the ethical values that underlie assessment processes, their relationship with social, environmental, political and cultural systems, and coherence with assessment practices. It includes learning about the role of assessment to address diversity, equity, socio-environmental justice and participation (removal of barriers), as well as the transposition between assessment and action research [6,7,9,16,21,25].

DIMENSION 2 (D2). Dimension of classroom assessment design: understood as the strategies designed in the classroom system consistent with the principles of EfS and IE, taking into account the relationship between the structure (what and how) and the function (why and what for) of assessment [46]. It is key to reflect on the assessment practices that are planned and developed, bearing in mind their current and future effects on all the elements that make up the assessment system. This reflection and activity must be guided by sustainable and inclusive values, against discrimination, exclusion and the deterioration of the environment [13,15,23,46].

After a first analysis of the data constructed, the different levels of complexity were established for each of the dimensions and the indicators describing them. The team of researchers used triangulation for the analysis system proposal, configuring it in the definitive analysis system (Table 2).

**Table 2.** Definitive analysis system.

| Level | D1: Socio-Environmental and Ethical Dimension | D2: Dimension of Classroom Assessment Design |
|:---:|---|---|
| 1 | They are not aware of the relation between the assessment system, its underlying values and its contribution to a sustainable and inclusive society. Their statements are inconsistent [1]. They understand that assessment should not be influenced by values in the interest of impartiality. They consider assessment as a verification and/or administrative requirement. They connect the ethical-moral, political-ideological dimension of assessment with political and government systems. | They do not know, or hardly know, the complexity of assessment processes. They do not define the kind of assessment. The design mainly focuses on the assessment tools. Assessment is to verify student knowledge, and the teacher is considered to be the only agent. They do not describe any relationship between structure (what and how) and function (why and for what). |
| 2 | They are aware of ethical aspects related to assessment (e.g., purposes [2] and reasons [3] for assessing), but not of the need for assessment systems to be based on inclusive and sustainable values (ethical positioning) as a response to the transformation of eco-social conflicts. They develop the concept of inclusive assessment through knowledge of IE, but not of the particularity of assessment from this perspective. | They know some types of assessment (formative, summative, initial, continuous and final assessment), but they do not recognise the regulating role of assessment. The consider some of the elements of the assessment system in the design, but without establishing relations between them. They do not establish relationships between the "what for" and "why" of each one of them. |
| 3 | They establish a relationship between assessment and integration of inclusive values and of EfS. There is coherence between how assessment practices are understood and how they should be developed from the perspective of sustainability and IE to achieve a fairer and more sustainable society. | They clearly set out the role of assessment as a mechanism to adjust the teaching–learning process. They consider the different elements in the design. They establish relationships between the "why" and "what for" of assessment (function) and the "what" and "how" (structure), and vice versa, and how these relationships condition the "when". They express and include principles of sustainability and inclusion. |

Note: [1] Incoherent and inconsistent repetition of certain knowledge acquired regarding assessment and other teaching–learning aspects. [2] Follow-up, learning and adaptation. [3] Know the impact of the educational activity, adjust the process, learn.

In the process of coding and analysis of the units of information, the NVivo 12 Plus software (QSR International Pty Ltd., Doncaster, Australia) was used, through which the coding matrices were obtained that allowed extracting frequencies and percentages, and organising the results.

Coding of the units of information was as follows:

S + n: information from a student (S). A number (from 1 to 24) was assigned to each of them.

G + n: information from a group (G). A number (from 1 to 5) was assigned to each of them.

The tool used was specified: IQ, SEP, pre-CAD, post-CAD, FQ and QEPK.

An example of final encoding would be: S1_FQ.

## 3. Results

The analyses of the tasks carried out show a total of 682 information units (henceforth UIs), which are presented in this section organised by dimensions (D) and levels (N). In each dimension, the results for the three phases of the described training process are included: initial phase, intermediate phase and final phase.

### 3.1. Socio-Environmental and Ethical Dimension

For this dimension, a total of 300 UIs were analysed.

In the initial phase, as shown in Table 3, 53.12% of the UIs are found in L1, and 42.7% in L2, which adds up to 95.82% in non-complex levels. These results reveal that most of

the students do not know the principles (values) and actions of an assessment from the perspective of EfS and IE. A superficial view of assessment processes is identified, basically limited to verifying student learning and to the classroom.

**Table 3.** Frequency and percentage of the different levels of classification defined for the socio-environmental and ethical dimension (Table 2) in each phase of the training process.

| Level | L1 | | L2 | | L3 | |
|---|---|---|---|---|---|---|
| | f | % | f | % | f | % |
| Initial phase | 49 | 51.04 | 41 | 42.70 | 6 | 6.25 |
| Intermediate phase | 13 | 9.15 | 39 | 27.46 | 90 | 63.38 |
| Final phase | 14 | 9.72 | 37 | 25.69 | 93 | 64.58 |

The students associate assessment with the need for impartiality. Most of them point out that the teacher is the only assessment agent, and that the teacher's values and ideas should not influence the assessment of the students. This is reflected in the UIs of S3_IQ: *In my opinion, assessing is not always done correctly, because we are not being entirely objective*, or in those of *S15_IQ: Both education and assessment are slightly biased, although the teacher tries to be impartial. Bias exists because of the teacher's previous experiences or thoughts.* S17_IQ: *In my opinion, it is difficult for assessment to be totally and completely neutral, although it should be so as much as possible.* However, there are students who put forward ideas related to assessment so as to improve student learning and other elements, such as S6_IQ: *I believe that assessment is necessary to know what aspects need to be improved in all areas.*

In this initial phase, no relationships are expressed between the values supporting the assessment from the perspective of EfS and IE, socio-environmental problems and the role of organising assessment systems in the eco-social transformation. In this phase, only one student, S13_IQ, expounds a comprehensive and complex view of assessment from this perspective:

*Assessment enables students to achieve better personal and academic development, and allows teachers to attain better personal and professional development. If it is carried out properly, it serves to improve the educational system at an institutional level (in regional and national educational centres). However, we could and should go one step further, as not only educational systems would improve. It would also serve for all of us to build a fairer and more sustainable society. Assessment is a non-linear tool in constant movement of adjustments, and all educational agents should participate in it. It should be considered as a tool of improvement for all (institutions, teachers, students, and families).*

The discourse of the rest of the students, despite having certain ideas and some knowledge related to a broader view of assessment than simply accreditation or students' grades, is incoherent. It is shown in the UIs of S14_IQ, who talks about changing the traditional approach assessment, but, at the same time, mentions it is necessary to check if the objectives set by the teachers have been reached: *I would not get rid of it* (assessment), *but I do think it should be changed to a less traditional approach. Assessment allows us to know if the objectives set by the teacher have been attained, and if the students have acquired the concepts and knowledge proposed during the teaching–learning period.* A lack of argument strength is observed in S20_IQ: *The main function of assessment is to measure if the students belonging to a certain context, exposed to a series of teaching–learning processes, have reached the minimum level established in achieving these processes with respect to specific content. As far as the area that encompasses the different components of the educational community is concerned, assessment acts as a continuous activity that provides constant feedback. This allows having an idea of what the students know about a specific topic.*

Likewise, in this initial phase, the students associate the ethical-moral and political-ideological aspects of assessment with governmental organisation and political parties, as can be seen in the UIs of S22_IQ: *Each educational centre and each nation has some specific political and ethical-moral ideals that influence teaching and, as a result, the process of assessing it.* Or, as S20_IQ comments: *Teaching in education is conditioned by the political decisions that are*

*made, and respond to the ideology of those who impose them.* S15_IQ mentions the following: *Every time there is a change of government in the country, educational laws are also modified in such a way that teaching and assessment are subject to these changes.*

Assessment aspects related to the organisation of the school, community resources, family participation, etc., are not included. Assessment is almost exclusively associated with verifying student learning. Most of the discourses revolve around it, as can be seen in the UIs of S10_IQ: *It can bring prestige or a bad reputation to the community and to the school, depending on whether the students' grades are good or bad, and on how the school assesses. It can cause happiness or sadness to families, depending on these same factors. For both teachers and students, it helps them to know if the student has acquired the knowledge dealt with during the teaching–learning process.*

In the intermediate phase, after implementing the values that support assessment from the perspective of EfS and IE in the classroom, and after the students performed the SEP task together, the number of students in L3 increased considerably. The UIs for L3 (Table 3) increased from 6.25% in the initial phase to 63.38% in this phase. The increase in the IUs for L3 occurred together with the decrease of IUs in L1 (now 9.15%) and L2 (now 27.46%). The data are shown in Table 3. Even in this phase, assessment from the perspective of IE was explained through the knowledge that the students had of it, without specifying the activities that refer to inclusive assessment, such as participation of students and other educational agents in decision-making in assessment processes, the design of authentic and inclusive assessment tasks, or the identification of barriers caused by excluding assessment processes.

In this phase, assessment is still considered as something to verify student learning, but more complex aspects are being included, as can be seen in the UIs of G3_SEP: *Assessment therefore serves to know, analyse, intervene in time, and improve the process. To carry out proper assessment, collecting any kind of data is worthless. The information that is useful to know how the teaching–learning process is developing is valid, not only by looking at the students, but at the entire educational community.*

In the final phase, 64.58% of the UIs were classified under L3. This level refers to a complex perception of assessment activities from the perspective of EfS and IE. Arguments against excluding assessment processes and how educational assessment systems should be organised to contribute to an eco-social transformation, and the development of critical and ethical thinking skills in the educational community were observed. This is how G5_SEP stated it: *This assessment is based on shared responsibility, a requirement or axis of democratic education and, therefore, inclusive and fair schools that allow us to achieve social justice*, or G3_SEP: *We consider that inclusive assessment, and therefore inclusive education, should create spaces in which children are forged as people capable of solving daily life problems, and that they do so from a perspective of democratic values that improve the society in which we live in a committed and free manner, a society that goes beyond work and production.*

In this phase, the idea of assessment as action research of the teaching practice, and the organisation of the school was observed. The students understand that assessment is a procedure that goes beyond verifying student learning to become a tool for adapting and investigating the educational activities designed and developed, to introduce improvements based on assessment and work together with all the agents involved. For example, G4_SEP mentioned the following: *An appropriate way to carry out teacher self-assessment is action research, which is a process that requires practical enquiry in a cooperative manner through action-reflection cycles. The objective of this practice is twofold: to improve the educational practice and turn it into something social, seeking the empowerment of the participants (. . . ) In addition, the socio-economic, social and cultural context in which the school is immersed will be taken into account. To this end, there will be collaboration with neighbourhood associations in order to work on the SDGs in an interdisciplinary and global manner, together with the rest of the curricular areas.*

The progression in the complexity of the students' ideas of assessment from the perspective of EfS and IE is related to how they rate the development of the course, as can be seen in the UIs of G2_SEP: *Several elements have benefited us, since it is one of the few courses in which the students were participants, both in the assessment, in the proposed activities, and the*

*specification of dates (. . . ) Through these activities, we were able to question ourselves regarding issues we had never thought of before, and they made us realise the need to adopt a new approach on how to conceive an assessment in accordance with the 21st century (. . . ) Furthermore, in this course, we were able to verify that assessment does not always have to be negative or unfair. The teacher let the students participate at all times in both the assessment and the activities that were going to be carried out. This is something unprecedented, since the majority of the teachers adopt an authoritarian attitude, where only and exclusively what they say is done (. . . ) Dialogue was used as the main element to know and share ideas about certain aspects of assessment that led to build our way of understanding assessment and the importance of organising the curriculum so as to offer inclusive education.*

In this final phase, data were also extracted from the questionnaire in which the students evaluated the progression of their knowledge (QEPK) of assessment from the perspective of EfS and IE.

In the first question about the students' evaluation of their level of knowledge constructed in the course, it was observed that the percentages of the answers were grouped into the three highest scores (Figure 3).

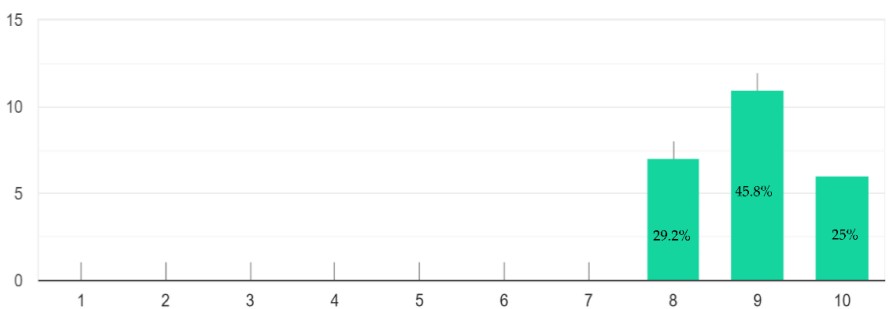

**Figure 3.** Percentages of responses in the students' evaluation of knowledge constructed in the developed training process.

With respect to the second question on rating the progression of the complexity of knowledge regarding assessment, the percentages were again grouped in the highest scores (Figure 4).

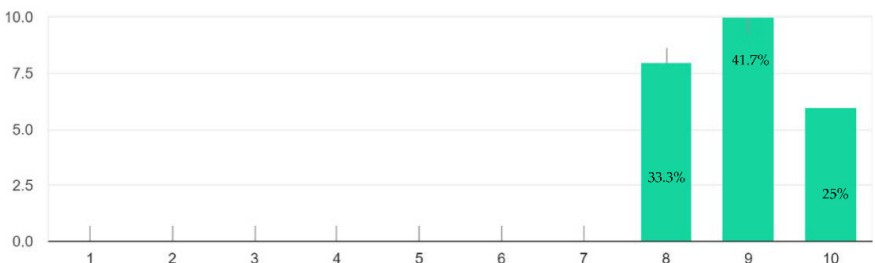

**Figure 4.** Percentages of responses in the students' evaluation of knowledge complexity constructed in the developed training process.

Regarding the specific questions referring to the socio-environmental and ethical dimension, the answers to questions 3 and 4 (Figure 2) were analysed. In the question about their knowledge of the ethical, environmental, political and social dimension of assessment, the percentages of responses were found at scores 6, 7, 8, 9 and 10. The highest percentage, 37.5% attained an 8. There are no answers for scores 1 to 5 (Table 4).

**Table 4.** Frequency and percentage of response when rating the knowledge built in the ethical, environmental, political and social dimension of the assessment.

| Score | Frequency | % |
|:---:|:---:|:---:|
| 6 | 1 | 4.2 |
| 7 | 6 | 25 |
| 8 | 9 | 37.5 |
| 9 | 6 | 25 |
| 10 | 2 | 8.3 |

When rating their knowledge of the inclusive dimension of assessment (community, attention to diversity, research, socio-environmental justice, participation) in question 4, the responses were again grouped in the highest levels, and there were no responses from 1 to 6 (Table 5). A score of 9 is the one that appears the most (45.8%).

**Table 5.** Frequency and percentage of response when rating the knowledge built in the inclusive dimension of assessment.

| Score | Frequency | % |
|:---:|:---:|:---:|
| 7 | 1 | 4.2 |
| 8 | 7 | 29.2 |
| 9 | 11 | 45.8 |
| 10 | 5 | 20.8 |

Regarding the qualitative evaluation in question 6 (Figure 2), it was observed that it was in tune with the previous scores, since most of the students explained how their knowledge of the socio-environmental and ethical dimension of assessment evolved in terms of quality and complexity. It can be seen in the following statements:

S5_QEPK: At first, I thought I had the necessary knowledge to deal with assessment as a teacher, but, after the course, I realised that my knowledge was very general and scarce. *Now, I consider that I have greatly evolved and have opened my field of vision much more with regard to assessment. There are concepts that I had not taken into account, others that I did not even know about, and still others that have shaken my initial ideas. In short, I have acquired numerous concepts, and have established a lot of relationships and connections with regard to assessment. An inclusive view is not only decisive in the methodology, it is also fundamental in assessment, because one cannot be understood without the other. I now know that assessment must be understood as a key part of the teaching–learning process. I also consider that it is of vital importance that we all participate in assessment and that we all assess ourselves.*

S24_QEPK: Really, thanks to everything I learnt, I understand assessment as the strategy for us to achieve social justice through an environmental aspect close to the students that arouses their interest and desire to investigate it. *We will thus also be able to bring them closer to the environment, and they will appreciate it, to have an impact on social aspects and local cultures to understand the global ones. In diversity lies the greatness of the human being. Institutions should participate in the context in the school, cooperating and collaborating, and citizens should act in accordance with their environment.*

*3.2. Dimension of the Classroom Assessment Design*

In this dimension, a total of 382 UIs were analysed. They provide us with valuable information on the progress of students' knowledge of the design of assessment activities carried out in the classroom from the perspective of EfS and IE. Table 6 shows the frequency and percentage of the different levels in the different phases of the training process.

**Table 6.** Frequency and percentage of the different levels of classification defined for the classroom assessment design dimension (Table 2) in each phase of the training process.

| Level | L1 | | L2 | | L3 | |
|---|---|---|---|---|---|---|
| | f | % | f | % | f | % |
| Initial phase | 51 | 53.12 | 31 | 32.29 | 14 | 14.58 |
| Intermediate phase | 13 | 31.7 | 25 | 60.97 | 3 | 7.31 |
| Final phase | 14 | 8.58 | 38 | 23.31 | 111 | 68.1 |

In the initial phase, the highest percentage is found in L1 (53.12%), although there is also a considerable percentage of students in L2 (32.29%). Only 14.58% is found in L3. These percentages reflect that the majority of students barely perceive the complexity of the design of assessment processes in the classroom, contemplating the values that support sustainable and inclusive assessment. They mainly focused on a type of final assessment carried out by the teacher, who is considered to be the only assessment agent. No relationships were established between the structure of the assessment, that is, what, when and how it is assessed, and its function: why and what for. This was observed in the following UIs:

S10_IQ: *We all have the ability to assess, but when it comes to the educational concept, at school, I believe that it should be carried out by professionals, that is, teachers.*

S14_IQ: *Thanks to assessment, we know if the objectives the teacher has set have been achieved and if the students have acquired the concepts and knowledge put forward during the teaching–learning period.*

S15_IQ: *The teacher is the one who decides what and how to assess in the classroom context based on the school's educational project.*

S20_IQ: *Assessment processes provide relevant and objective information on the students' knowledge.*

More complex knowledge, characteristic of L2, was observed in some students. They contemplated different types of assessment, but again, only focused on verifying if the students achieved the objectives established.

S1_IQ: *Systematic and formative assessment allow reporting on the process from the beginning to the end, so that students know if they are progressing or regressing.*

The regulating role of assessment in the different elements of the process is not recognised: students, teachers, methodology, materials, tasks, tools, family, school, social context, etc. Examples are shown in the following statements:

S14_IQ: *Teachers use it to assess their teaching activity and to know the problems certain students may have with some content that is being taught and the way of teaching it.*

S23_IQ: *It allows observing the needs of every student, how they learn best, what their interests are, where teaching practice fails, etc.*

It was also observed that some students, although a low percentage, included elements of the assessment system: the how and what, but without being aware of the relationship with its function. It can be seen in the following UIs:

S6_IQ: *There are many aspects that can be assessed not only on the basis of learning. The materials used, the teaching methods, how the activity has been developed,* etc. *can be assessed.*

S17_IQ: *I think it would be more appropriate, after knowing the group and after reaching an agreement, to choose how to assess. I believe that assessment should not be imposed, and especially that assessment tools should help the teacher in this assessment process.*

In the intermediate phase, the percentages corresponding to L2 were the majority (60.97%). The IUs referring to L1 dropped to 31.7%. The different types and moments of assessment, and objects of assessment different to the students' knowledge were taken into account. This progression in the students' ideas was observed in the following UIs:

G1_pre-CAD: *Assessment will be present throughout the educational process, because we start from the idea that every educational practice entails an assessment process. For this reason, our assessment will be a systematic process (at the beginning, during the educational process and at the end).*

G4_pre-CAD: *At the end of every session, the students can freely write their sensations, experiences, feelings, concerns and barriers or needs experienced during the session in the teaching journal. We thus let the students participate in the assessment process, giving them voice. Teachers can use this tool as self-assessment and it enables them to see how their students are doing.*

Still in the intermediate phase, a mere 7.31%—and only in some of the UIs of group 3—is found in L3, as shown by G3_pre-CAD: *All this gives meaning to a social assessment that allows identifying how progress is being made in order to identify what needs to be modified (…) These tools help us perform work based on feedback with the students to know where we are and where we are going (…) It is necessary to adjust each tool to the nature of the activity. Observation, evidence of the process and a rubric are considered to be amongst them.*

In the final phase, however, the percentage of UIs increased in L3 (68.1%). In this phase, 23.31% of the students attained L2 and only 8.58% remained in L1. The students frequently explained the role of assessment to adapt the teaching–learning process and mentioned the current and future effects of assessment activities. They know about the different elements of the assessment system and how those elements are related in a complex way. They also connected classroom assessment design with school systems and with the social context, explicitly including principles of sustainability and inclusion. The UIs coded in the tools of this phase illustrate the pre-service teachers' progress:

G1_post-CAD: *This assessment will be carried out systematically at the beginning, during and at the end by means of action research (…) Based on the previous ideas, it should not be forgotten that the view of the teachers regarding assessment, as well as their ethical principles and experiences, will influence the assessment system, since ethics itself configures the methodological system applied in the classroom.*

G2_post-CAD: *In short, it is observed that all these aspects are interrelated, because if we alter or change any of them, the entire system would be modified and it would have to be readjusted to the needs or demands of the courses, objects of study, etc.*

G3_post-CAD: *Assessment is a key tool that helps reflecting on and improving the student-teacher relationship (…) An assessment closely linked to ethics and sustainability, understood as the process of training critical and democratic citizenship from an ethical and active perspective the objective of which is the development of societies that act for a better world, that go from the local to the global, and that evolve from individual to collective arguments (…) the importance of the process, observation, feedback, sharing, decision-making, the variety of tools, individuality and community, reflection, negotiation, etc. Furthermore, during these days, we have left behind criteria such as impartiality or the neutral sense of assessment (…) The regulating function of assessment allows us to analyse the responses of the students, if we focus on identifying the possible difficulties (…) it allows adjusting the process (modification of previous ideas, evaluation progression and introduction of error as a source of learning).*

As pointed out earlier, in this last phase, data were also extracted from the questionnaire in which students evaluated the progression of their knowledge (QEPK) of assessment from the perspective of EfS and IE. In this dimension, data from the answers to question 5 (Figure 2) were included: "Rate your level of knowledge of the dimension of classroom assessment design on a scale from 1 to 10". The scores obtained are all 7, 8 and 9 (Table 7).

**Table 7.** Frequency and percentage of response in the evaluation of the knowledge built of the dimension of classroom assessment design.

| Score | Frequency | % |
|:-----:|:---------:|:----:|
| 7 | 3 | 12.5 |
| 8 | 11 | 45.8 |
| 9 | 10 | 41.7 |

Regarding the qualitative evaluation of question 6 (Table 2), the students indicated the opportunities provided in the course for their knowledge of the classroom assessment design to evolve from a sustainable and inclusive perspective. They highlighted the devel-

oped capacity to integrate theoretical principles with specific activities in the classroom. The following statements clearly show this:

S11_QEPK: *My knowledge of assessment has evolved positively during the course. At first, my ideas about assessment were related to the tools and their design in a classroom. In this regard, during the education degree, I did not know the meaning of assessment regarding why and what to evaluate. And even more so, I did not know either about assessment from the perspective of ethics, sustainability and inclusive education.*

S17_QEPK: *At the beginning of the course, I had some knowledge of educational assessment. However, throughout the course, I understood new terms and parts of the assessment process (which are sometimes considered as learnt, but are not). An example of this is the identification of the object of assessment and the suitability of criteria and tools. That is, what, how, when and why to assess (...) Finally, re-drafting a design was complex, both at a personal level and at group level, since it was the moment to apply the content worked on and learnt. Honestly, doing this last group activity implied reorganising ideas and beliefs regarding assessment.*

Finally, it should be noted that in this general qualitative assessment of the development of knowledge, the students highlighted that implementing those self-same sustainable and inclusive values that they intended to build in the course allowed them to experiment and integrate them:

S11_QEPK: *Moreover, in this course, I have been given the opportunity to start from my previous knowledge by developing a script of initial questions. In fact, the possibility of doing this at the end of the course invited us to know how our knowledge construction had evolved. Also, teamwork has brought with it a process of knowledge restructuring and constructive learning. The possibility of carrying out the work at first without any knowledge of the course, and reformulating it again at the end of the course, in my case, made it possible for me to know that the world of assessment is much broader than I thought, and is not only limited to what we study in the degree. Likewise, the dialogues in class with the teachers, the teamwork and reading a wide variety of bibliographic documents, enriched knowledge and, they enriched me both personally and professionally. In short, I am highly satisfied both with the management and organisation of the course, and with the dynamics carried out in class (writing during the dialogue between the teacher and the students in small practical classes, developing an assessment tool on the work, and the oral presentation in large groups). Last but not least, the willingness and help of both teachers towards the students showed their high degree of empathy at all times. THANK YOU!*

S12_QEPK: *There were aspects of the assessment that I had never questioned and, thanks to this course, I have had the opportunity to reflect on it in depth. A need has arisen in me to enquire about assessment and the aspects it encompasses.*

S22_QEPK: *I consider that, thanks to the explanations and tasks performed during the course, I have been able to build my knowledge of assessment in a more complex way.*

## 4. Discussion

The discussion and synthesis of the results following the research objectives are explained hereafter.

### 4.1. Analysis of the Progression of Pre-Service Teachers' Ideas and Construction of Knowledge of the Socio-Environmental and Ethical Dimension That Affects Assessment Systems

With respect to the first research objective, an increase in the complexity of the students' ideas was observed as the training advanced, which went from categorising almost all of the UIs under non-complex levels (L1 and L2) in the initial phase to classifying more than half of them in the most complex level (L3) in the final phase. The progression of these ideas was tinged with superficial arguments related to the assessment processes, the verification of student learning, or the need for impartiality. According to Flores [47], people are consumers of already formulated scientific ideas. They include aspects such as opinions, beliefs, perceptions and conceptions. With regard to the ideological aspects of assessment, its understanding and implementation allow revealing aspects related to the nature of intelligence, the teaching–learning process, the teaching profession itself, the structure and

social dynamics, and the purpose of institutions, amongst others [48]. This philosophical-scientific paradigm is consistent with the structures of advanced capitalism [49].

At the end of the course, the students' ideas became more complex. They considered assessment as a tool to adapt and investigate the educational activities carried out and to introduce improvements together with all the agents involved. It was observed that the students advanced towards assuming assessment as decisive for all other activities associated with the intervention and, of course, with learning [13,50]. They aligned assessment with improving the quality of the training activity [17,46] as soon as they perceived it as an indicator to adapt the teaching–learning processes to the needs of the students, the teachers and the process itself. These are all included aspects in the theoretical framework that indicate that assessment systems were understood from more complex views.

Furthermore, they rejected exclusion in assessment processes and mentioned how educational assessment systems should be organised to contribute to eco-social transformation, and to the development of critical and ethical thinking skills in the education community. They were aware of how their own ideas evolved and recognised the influence that the subject of linking assessment with EfS and IE has had on this progression, not only with regard to the knowledge built, but also to the ethical, environmental, political and social dimension, or the inclusive dimension of assessment.

It would be interesting to analyse to what extent these changes are implemented in the assessment designs of the students in future research and to be able to monitor their future professional practice.

*4.2. Analysis of the Progression of Pre-Service Teachers' Ideas and Construction of Knowledge of the Classroom Assessment Design from the Perspective of EfS and IE*

As in the previous research objective, the ideas of the students in the initial phase were mainly categorised in non-complex levels. As said earlier, the students referred to assessment as something exclusive to the teacher and to verify student learning at the end of a course. Relationships were not usually established between the structure and the function of this assessment, nor was there any conscious allusion to the different elements: how, what, when, why and what for. Although other types of assessment were identified, none of the students identified its regulating function.

As the training process advanced, ideas and knowledge evolved towards more complex levels. In the final phase, more than half of the UIs analysed were found in the highest level of complexity. In this level (L3), the students specified the elements of the assessment system and how they are related to each other, an aspect that provided the systemic behaviour of assessment in the classroom with dynamism and complexity [46]. Moreover, they gave importance to the regulating function of assessment and its effects. This issue is of great relevance in studies of this kind that seek to provide students, citizens and future teachers with an inclusive and sustainability perspective as a moral support. Assessment should be understood in its context and the data gathered through assessment do not only adjust the process, but also the actions of the teacher and the students themselves with the intention of improving the quality of the training activity [17,46].

The students recognised the influence of the subject in the progression of their ideas, in addition to identifying how and to what extent these ideas evolved with regard to the classroom assessment design from a sustainable and inclusive perspective. This is highly encouraging, since it reflects student empowerment [11,13] from appropriating meanings in their own discourses. Assessment has also turned into an element of learning [19].

In this regard, it is important for the students to emphasise the developed capacity to integrate the ethical and theoretical principles addressed in the training process with specific activities in the classroom system. This refers to the effectiveness of the training process, which includes the same sustainable and inclusive values the students are expected to build. As educators of future teachers, this aspect is of vital importance, as, in our opinion, there should be coherence between the assessment processes implemented and the educational model we defend, that is, the medium as the message [46].

## 5. Conclusions

### 5.1. General Conclusions

From the analysed data, it is inferred that this kind of subject responds to the challenge we face as teachers of questioning the prevailing negative values of global problems through the complex view of sustainability and IE. The results obtained clearly show EfS and IE have acted as guiding frameworks in the ideological and ethical dimension of training future teachers [9].

In light of the above, it is considered that the perspectives of EfS and IE provided support, both in theory and in practice, and served not only to design assessment processes, but the course as a whole. As expected, they were intentionally and consistently included in the designs of the assessment systems developed by the students, which is observed in their considerations.

To conclude, based on the data analysed, it can be said that the developed training process has had a positive impact on the students' ideas and knowledge regarding assessment from the perspective of EfS and IE.

### 5.2. Limitations of the Study and Future Research

Both the training proposal and the study itself have limitations, which will serve as a starting point for further improvements. On the one hand, the duration of the training process was limited and a greater number of sessions would have allowed us to delve deeper into aspects that require a lot of dialogue and debate (planetary ethics). Restructuring previously acquired conceptions of people is a complex process that requires time. On the other hand, another limitation is not having carried out an impact assessment through which to observe if restructuring ideas and the progression of complex theoretical-practical knowledge of assessment from the perspective of EfS and IE are maintained over time and are integrated into the students' professional performance.

It has not been possible to carry out an impact assessment because not enough time has passed since the training process was implemented.

This proposal is related to a context from which ideas can be extracted to be transferred to other similar scenarios or groups. Its principles and structure can serve as a reference that can be adapted to the different training contexts in which it can be put into practice. It will, therefore, be necessary for the teaching teams to integrate the principles of assessment from EfS and IE into their educational processes.

In order to improve, the training process would have to be implemented again. The necessary readjustments can be made once an entire cycle has been completed (design, implementation, research, dissemination). Furthermore, as said earlier, it would be interesting to carry out an exhaustive analysis of the students' assessment system designs and to monitor their future professional practice.

All these aspects could have a positive impact on the intervention proposal presented and, therefore, on EfS and IE.

**Author Contributions:** Conceptualization, B.G.-N., R.J.-F. and P.A.; methodology, B.G.-N., P.A. and E.G.-G.; software, B.G.-N.; validation, E.G.-G., R.J.-F. and P.A.; formal analysis, B.G.-N. and P.A.; investigation, B.G.-N., E.G.-G., R.J.-F. and P.A.; resources, E.G.-G.; data curation, E.G.-G.; writing—original draft preparation, B.G.-N., E.G.-G. and P.A.; writing—review and editing, B.G.-N., E.G.-G. and R.J.-F.; visualization, E.G.-G. and R.J.-F.; supervision, P.A.; project administration, B.G.-N. and P.A. All authors have read and agreed to the published version of the manuscript.

**Funding:** This research received no external funding.

**Institutional Review Board Statement:** Not applicable.

**Informed Consent Statement:** Informed consent was obtained from all subjects involved in the study.

**Data Availability Statement:** The datasets presented in this article are not readily available. Requests to access the datasets should be directed to beatriz.gallego@uca.es.

**Acknowledgments:** The authors gratefully acknowledge the support of the "Desarrollo Profesional del docente" Research Group of the Universidad de Cádiz.

**Conflicts of Interest:** The authors declare no conflict of interest.

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
