# Peer review of "Inclusive Assessment and Sustainability Dimensions: Pre-Service Teachers’ Ideas and Knowledge"

_sustainability, doi:10.3390/su14159651_

Round 1

Reviewer 1 Report

Dear Authors:

I congratulate you on the very interesting work you have presented. It is a coherent and rigorous study that really provides relevant and interesting results that broaden the knowledge about teacher education with respect to assessment, within a novel framework. Below, I leave some comments and suggestions that could help to further strengthen the manuscript:

1. In the abstract briefly discuss the results obtained from the EfS point of view. I think that adding the results also from the IE point of view would also be interesting.

2. Although the relationship between EfS and IE is clear from the statements made in SDG 4, I think it would be useful to add a paragraph reflecting on this relationship from the more specific point of view of the authors. In other words, the coherence between both paradigms is clear, but why do the authors choose these axes and not others to structure their ethical framework?

3. Both in the introduction and in the conclusions, it would be interesting to include a brief explanation of the potential of this study for the field of science education in general.

4. I would change the title "Materials and Methods" to "Methods", which is more appropriate for this study, since the materials are already explained in a subsection.

5. In the subsection "2.1. Research context and participants" an explanation of the phases has been included. However, these phases should be included in another subsection called, for example, procedure or data collection.

6. It would be convenient to specify briefly what type of adaptation of the Individual initial questionnaire (IQ) has been made. It is also necessary to briefly indicate the reliability and validity of the other instruments used.

7. It is necessary that the authors do not abuse so much of self-citations and that they eliminate some of them, replacing them with other important references from the literature.

Once again, congratulations and I hope that these comments will be useful in shaping the final version of the manuscript.

Author Response

Thank you very much for your kind words and encouraging comments.

We greatly appreciate your efforts. We consider your contributions have helped to considerably improve the quality of our paper.

Below we present our responses to the different comments.

  1. In the abstract briefly discuss the results obtained from the EfS point of view. I think that adding the results also from the IE point of view would also be interesting.

Thank you for pointing this out. We have added the results obtained from the perspectives of EfS and IE.

  1. Although the relationship between EfS and IE is clear from the statements made in SDG 4, I think it would be useful to add a paragraph reflecting on this relationship from the more specific point of view of the authors. In other words, the coherence between both paradigms is clear, but why do the authors choose these axes and not others to structure their ethical framework?

We have added a paragraph in which we explain this issue. The arguments have been supported by specific literature.

  1. Both in the introduction and in the conclusions, it would be interesting to include a brief explanation of the potential of this study for the field of science education in general.

The introduction already explains, to a certain extent, the potential of the study for the training of future teachers. The idea has been reinforced, adding the potential for educational sciences in general. We consider that the approach of this idea in the theoretical foundation improves the quality of the discourse. However, we believe that including it in the conclusions may seem repetitive.

  1. I would change the title "Materials and Methods" to "Methods", which is more appropriate for this study, since the materials are already explained in a subsection.

We have made this change.

  1. In the subsection "2.1. Research context and participants" an explanation of the phases has been included. However, these phases should be included in another subsection called, for example, procedure or data collection.

We have added another subsection.

  1. It would be convenient to specify briefly what type of adaptation of the Individual initial questionnaire (IQ) has been made. It is also necessary to briefly indicate the reliability and validity of the other instruments used.

The adaptation made has been specified in the text.

With regard to the reliability and validity of the instruments, we think that, as it is a qualitative study and action-research, these are not considered from statistical procedures, but the criteria of credibility, confirmability and transferability are ensured through reflective processes on behalf of the teaching and research team (teachers-students). This has been explained in the text.

  1. It is necessary that the authors do not abuse so much of self-citations and that they eliminate some of them, replacing them with other important references from the literature.

We have eliminated some of the self-citations. New references have been added to support the text.

Once again, congratulations and I hope that these comments will be useful in shaping the final version of the manuscript.

Thank you very much for your comments.

Reviewer 2 Report

This study presents the results of a training course on assessment in the perspective of Sustainability Education and Inclusive Education, aimed at pre-service teachers. 

This is a predominantly qualitative study that aims to evaluate the effectiveness of the aforementioned course. The work is well structured, the theoretical framework is clearly explained, the objectives of the study appear consistent with the theoretical premises. The authors appropriately reported the two main limitations of the study.

Suggestions for authors

p.2. When assessment is understood as (self) regulation [16], and as an element of learning [17] and empowerment [9,12], it acquires a determining role in the configuration of the ethical dimension of students, future professionals, teachers, and citizens. 

With reference to the aforementioned statement, authors should also refer to the relevance of the feedback provided by the assessment for the development of academic self-efficacy (e.g., Schunk & DiBenedetto, 2014; 2022).

p.4. It was a participatory process based on a constructivist, socio-affective and critical approach, which was structured into three phases that combined individual and group work:

With reference to the aforementioned statement, the authors should report some references in support of the chosen methodology.

p.6 Was the questionnaire for assessing the progression of knowledge (QEPK) built ad hoc or is it a standardized questionnaire?

In the section Discussion and conclusions (4) it would be advisable for the authors to discuss the strengths and weaknesses related to the implementation of the intervention and provide guidelines on the possible generalizability of the proposed training model.

Author Response

Thank you very much for your comments.

We greatly appreciate your efforts. We consider your contributions have helped to considerably improve the quality of our paper.

Below we present our responses to the different comments.

Suggestions for authors

p.2. When assessment is understood as (self) regulation [16], and as an element of learning [17] and empowerment [9,12], it acquires a determining role in the configuration of the ethical dimension of students, future professionals, teachers, and citizens.

With reference to the aforementioned statement, authors should also refer to the relevance of the feedback provided by the assessment for the development of academic self-efficacy (e.g., Schunk & DiBenedetto, 2014; 2022).

Thank you for pointing this out. We have included the references in the text.

p.4. It was a participatory process based on a constructivist, socio-affective and critical approach, which was structured into three phases that combined individual and group work:

With reference to the aforementioned statement, the authors should report some references in support of the chosen methodology.

We have included references to support the idea.

p.6 Was the questionnaire for assessing the progression of knowledge (QEPK) built ad hoc or is it a standardized questionnaire?

In the text, we have explained the procedure followed.

In the section Discussion and conclusions (4) it would be advisable for the authors to discuss the strengths and weaknesses related to the implementation of the intervention and provide guidelines on the possible generalizability of the proposed training model.

We have included your suggestions.

Reviewer 3 Report

The article: INCLUSIVE ASSESSMENT FROM THE PERSPECTIVE OF SUSTAINABILITY: PRE-SERVICE TEACHERS’ IDEAS AND KNOWLEDGE It brings to the table a little-known and little-worked topic of great interest to the educational sciences and society in general. The link they make between sustainability, equity and assessment methods for teaching and learning process is really interesting and innovative. Well conducted and strengthened, this article could be a great contribution to help the education community to rethink not only their forms of evaluation, but also to understand the importance school evaluation systems may have in achieving sustainability and social justice. In doing so, some important aspects need to be improved in the current version of the manuscript. 

The first one I would like to express is the need for greater clarity and concreteness in the way the article is written. This includes: the need for a thorough editing process in English, which often makes it difficult to understand what is meant, and the need to explain some key aspects of the paper with more clarity and references, especially regarding the methodology used in your research.

Bellow I include some examples of sentences that need English editing or content reformulation, with proposals of change in some cases:

L6 in the abstract: To collect data, a variety of tools, amongst which the students’ group work, …-> Here the meaning is not clear

L15 in the introduction section: “Particularly related to our study is target 4.7: ensure that all learners acquire the knowledge and skills needed to promote sustainability, including, amongst others, through EfS and sustainable lifestyles, human rights, gender equality, promotion of a culture of peace and non-vio- lence, global citizenship and appreciation of cultural diversity and of culture’s contribu- tion to sustainability.” -> it is a very long sentence in English and the meaning ma be misunderstood. Please revise it. At least it should go like this: “Particularly related to our study is target 4.7: ensure that all learners acquire the knowledge and skills needed to promote sustainability, including through EfS and sustainable lifestyles: human rights, gender equality, promotion of a culture of peace and non-violence, global citizenship and appreciation of cultural diversity and of culture’s contribution to sustainability.”

L27 in the introduction section: “critical of situations of discrimination and inequality,… “ are you talking about critical thinking? Please specify. 

L34 in the introduction section: change “the capacity to act in the development of our society” for a more specific sentence that could be something like: the capacity to act in favour of a more sustainable and fair development.

L 36: change “and how want to train them in the future.“ for “and how we want to train them in the future...“ I suggest to start the sentence with something like: The implemented didactic intervention consisted of…

- Along the test, you use “said educational model” “said assessment...”, it does not seem English like. I suggest revising such sentences. For example in L44 it is better to say express your idea as follows: “…plays a key role in achieving this educational model”.

- Along the test, please pay attention to the use of the word subject. For the course you are analysing, please, use the word course. For the subject itself, this is, the linkage between assessment and EfS and IE, then use the word subject specifying it the first time you use it what you mean with the use of the word subject.

METHODOLOGICAL ASPECTS ON THE MANUSCRIPT:

In the introduction section you state: “Starting from the review of the literature, a series of principles are developed to guide us in the design and implementation of assessment around” It is not clear what you mean and specially what where the steps you followed, so it would be very important if you could clarify. 

In the first paragraph of materials and methods, you say “To this end, two funda- mental spheres of study were addressed throughout the process: 1) the socio-environmen- tal and ethical dimension that affects assessment systems, and 2) the classroom assessment design from the perspective of EfS and IE. Afterwards, it you should clarify what you mean with each, and provide reference if possible. 

When you expain the intermediate phase, in point 2 you say “ased on an initial design. After introducing changes to the initial design, …” But you do not expecify who this process was done. Do students first work in groups on the first task and then work together as a whole class and share what they have worked on in small groups and try to find consensus? Please, opacify so in the methodological section.

Tables and figures need to stand alone, this is, the footnote to a table or figure should have enough information to be able to understand and interpret them. Please, explain clearly what each table do represent: example given, table 6 could be changed to: Frequency and percentage of the different levels of classification defined for the classroom assessment design dimension (table 2) in each phase of the training process. 

Or: Table 3: Results of classicisation of Units of information (UIs) for the socio-environmental and ethical dimension according to.

You say: a class journal was used as a tool to assess the process -> you mean the teaching and learning process, the design of the didactic intervention or the research process? Please try to be more concrete. 

Also, some aspects of the methodologies implemented to raise conclusions in these research are not clear. For example, In figure 2 you do not say that you asked to justify their answers, but them it the results, it seems you did.

Some sentences need also further explanation of better linkage to undersand what you did. For example, these sentences need further explanation:

Report of the theoretical-practical assessment proposal from the perspective of EfS and IE for a school (school-based educational project, or SEP), prepared in the intermedi- ate phase 

Classroom assessment design from the perspective of EfS and IE (CAD) 

It is not clear what you evaluated and how. 

Similarly, you state: “After a first analysis of the data constructed, “ but you did not specify what this analysis is based on.

This sentence is found in the results section but would be better in the metholody section: dents evaluated the progression of their knowledge (QEPK) of assessment from the per- spective of EfS and IE. The questionnaire consisted of 5 closed-ended questions in which the students had to rate different aspects of the knowledge worked on in the subject from 1 (not achieved) to 10 (fully achieved) and one open-ended question (Figure 2). A total of 24 students responded. 

Please explain how you grouped results in table 4 and table 5.

As expressed above, some aspects of the methodology used need to be clearer explained in order to better interpret your results. Specially, the way you took information on the process during the intermediate phase is not clearly explained.  

DISCUSSION

4.1.

You state that “The progression of these ideas was tinged with superficial arguments related to the assessment processes, the verification of student learning, or the need for impartiality”. If it is a progression, such progression should be noted in your statement. This is, students ideas started with superficial ideas and along the process the complexity of argumentation was detected and so on.

This sentence needs to be revised in terms of the way it is written: It is clear that EfS and IE have acted as guiding frameworks for the ideological and ethical dimension of the training of the students, future teachers 

Still, as I read it, it occurs to me that you could mention here projects as teachers for the future, and include as well as other references. It would also be important to mention the need for future research that you have identified.

4.2

In sentence “As in the previous research objective, the ideas of the students in the initial phase were mainly found in non-complex levels” I suggest you change the word "found" to "categorised", as what you have done is a categorisation of the ideas of the students.

In sentence “This question is of great relevance in studies of this kind …”change “this question” by “this issue is of great relevance…”

Regarding paragraph: “In light of the above, it is considered that the perspectives of EfS and IE provided moral support, both in theory and in practice, and served not only to design assessment processes, but the subject as a whole. As expected, they were intentionally and consistently included in the designs of the assessment systems developed by the students, which is observed in their considerations.” First of all, I would like to point out that it serves as an example of the need for the article to distinguish between your course or teaching intervention and the topic you are proposing. This is a matter of linguistic editing, but it is important that you handle it to make your work clearer. Secondly, when you refer to "their considerations" you have to relate it to the assessment methods you have used in your research. That is, you have to refer to the evidence you have to draw conclusions. 

CONCLUSIONS

I would suggest you to include a separate section with your general conclusions, limitations of the study and proposals for future research. 

Here I suggest you to reinforce your current conclusions. In doing so, I propose you to avoid sentences as: “we can confirm” and to make statements more adjusted to what you actually have done: “Based on our results obtained… “ and here I proposes (but please adjust it to your real circumstances) “in this first didactic intervention focussed on the perspective of EfS and IE in the design and development of teaching assessments to solve the problems associated to Assessment…”

On regard to the limitations of the study, I would propose to consider to mention that this has been a pilot implementation of your intervention, and that this analysis will serve you to validate some aspects of it and improve others. 

Besides, in your current conclusions you say “In other words,…” but what you state is a new idea. So I suggest you restructure your conclusions reinforcing each of the two ideas you are defending. Furthermore, I think you should include in your conclusions and future research section the idea that this paper is “a contribution to help the education community to rethink not only their forms of evaluation, but also to understand the importance school evaluation systems may have in achieving sustainability and social justice” and point out further steps and research that are needed to continue working in that direction. 

REFERENCES

In the case of non-English references, which are abundant in this paper, please provide a translation into English so readers may understand the content of such reference. 

Apart from the link you include, please provide in all cases the name of the journal where it is published (e.g. In these references it is missing: 15, 16, 20, 21, 25, 28) an further information when needed (e.g. 17).

Thank you, I am willing to read your improved version of the manuscript. 

Author Response

Thank you very much for your comments.

We greatly appreciate your efforts. We consider your contributions have helped to considerably improve the quality of our paper.

Below we present our responses to the different comments.

Comments and Suggestions for Authors

The article: INCLUSIVE ASSESSMENT FROM THE PERSPECTIVE OF SUSTAINABILITY: PRE-SERVICE TEACHERS’ IDEAS AND KNOWLEDGE It brings to the table a little-known and little-worked topic of great interest to the educational sciences and society in general. The link they make between sustainability, equity and assessment methods for teaching and learning process is really interesting and innovative. Well conducted and strengthened, this article could be a great contribution to help the education community to rethink not only their forms of evaluation, but also to understand the importance school evaluation systems may have in achieving sustainability and social justice. In doing so, some important aspects need to be improved in the current version of the manuscript. 

The first one I would like to express is the need for greater clarity and concreteness in the way the article is written. This includes: the need for a thorough editing process in English, which often makes it difficult to understand what is meant, and the need to explain some key aspects of the paper with more clarity and references, especially regarding the methodology used in your research.

A native English speaker specialised in editing scientific papers has revised our paper. We trust the modifications made based on your comments enabled solving the questions raised.

Below I include some examples of sentences that need English editing or content reformulation, with proposals of change in some cases:

L6 in the abstract: To collect data, a variety of tools, amongst which the students’ group work, …-> Here the meaning is not clear

L15 in the introduction section: “Particularly related to our study is target 4.7: ensure that all learners acquire the knowledge and skills needed to promote sustainability, including, amongst others, through EfS and sustainable lifestyles, human rights, gender equality, promotion of a culture of peace and non-violence, global citizenship and appreciation of cultural diversity and of culture’s contribution to sustainability.” -> it is a very long sentence in English and the meaning may be misunderstood. Please revise it. At least it should go like this: “Particularly related to our study is target 4.7: ensure that all learners acquire the knowledge and skills needed to promote sustainability, including through EfS and sustainable lifestyles: human rights, gender equality, promotion of a culture of peace and non-violence, global citizenship and appreciation of cultural diversity and of culture’s contribution to sustainability.”

L27 in the introduction section: “critical of situations of discrimination and inequality, … “ are you talking about critical thinking? Please specify. 

L34 in the introduction section: change “the capacity to act in the development of our society” for a more specific sentence that could be something like: the capacity to act in favour of a more sustainable and fair development.

L 36: change “and how want to train them in the future“ for “and how we want to train them in the future...“ I suggest to start the sentence with something like: The implemented didactic intervention consisted of…

- Along the test, you use “said educational model” “said assessment...”, it does not seem English like. I suggest revising such sentences. For example, in L44 it is better to say express your idea as follows: “…plays a key role in achieving this educational model”.

- Along the test, please pay attention to the use of the word subject. For the course you are analysing, please, use the word course. For the subject itself, this is, the linkage between assessment and EfS and IE, then use the word subject specifying it the first time you use it what you mean with the use of the word subject.

METHODOLOGICAL ASPECTS ON THE MANUSCRIPT:

In the introduction section you state: “Starting from the review of the literature, a series of principles are developed to guide us in the design and implementation of assessment around” It is not clear what you mean and specially what where the steps you followed, so it would be very important if you could clarify. 

A set of principles are built from the review of the literature on assessment, IE and EfS with the aim of having a theoretical basis to design assessment from the perspective presented here. The steps followed for the literature review are those normally followed in research processes (establish search criteria, search in the main databases: ERIC, EDUCATION DATABASE, SCOPUS, WEB OF SCIENCE, etc., and selection and analysis of the most suitable articles). We consider it is not convenient to explain them in the theoretical foundation section.

In the first paragraph of materials and methods, you say “To this end, two fundamental spheres of study were addressed throughout the process: 1) the socio-environmental and ethical dimension that affects assessment systems, and 2) the classroom assessment design from the perspective of EfS and IE. Afterwards, you should clarify what you mean with each, and provide reference if possible. 

These dimensions have been constructed based on the rationale presented in the introduction, and are explained on page 8. However, some of these references have been included.

DIMENSION 1 (D1). Socio-environmental and ethical dimension: understood as the awareness of the ethical values that underlie assessment processes, their relationship with social, environmental, political and cultural systems, and coherence with assessment practices. It includes learning about the role of assessment to address diversity, equity, socio-environmental justice, and participation (removal of barriers), as well as the transposition between assessment and action research [6, 7, 9, 16, 23, 27].

DIMENSION 2 (D2). Dimension of classroom assessment design: understood as the strategies designed in the classroom system consistent with the principles of EfS and IE, taking into account the relationship between the structure (what and how) and the function (why and what for) of assessment [32]. It is key to reflect on the assessment practices that are planned and developed, bearing in mind their current and future effects on all the elements that make up the assessment system. This reflection and activity must be guided by sustainable and inclusive values, against discrimination, exclusion and the deterioration of the environment [13, 15, 25, 45]

When you explain the intermediate phase, in point 2 you say “based on an initial design. After introducing changes to the initial design, …” But you do not expecify who this process was done. Do students first work in groups on the first task and then work together as a whole class and share what they have worked on in small groups and try to find consensus? Please, opacify so in the methodological section.

It has been clarified in the text.

Tables and figures need to stand alone, this is, the footnote to a table or figure should have enough information to be able to understand and interpret them. Please, explain clearly what each table do represent: example given, table 6 could be changed to: Frequency and percentage of the different levels of classification defined for the classroom assessment design dimension (table 2) in each phase of the training process. 

Or: Table 3: Results of classicisation of Units of information (UIs) for the socio-environmental and ethical dimension

We have made the changes recommended for Table 6, and have adapted the caption for Table 3.

You say: a class journal was used as a tool to assess the process -> you mean the teaching and learning process, the design of the didactic intervention or the research process? Please try to be more concrete. 

We refer to the teaching journal. It is an assessment and reflection instrument for teachers, by means of which they gather information on the development of the dynamics in the classroom, the learning process of the students, their teaching practice and even on the didactic design. It was used as yet another instrument for collecting data, as specified in the methodology section. It was also used to adapt the teaching-learning process.

Also, some aspects of the methodologies implemented to raise conclusions in these research are not clear. For example, in figure 2, you do not say that you asked to justify their answers, but them it the results, it seems you did.

The students were asked to provide a greater explanation of the answers expressed in questions 1-5 in question 6: “Rate how the complexity of your knowledge of assessment has evolved in a qualitative manner during the course on a scale from 1 to 10.” These explanations are part of the results.

Some sentences need also further explanation of better linkage to understand what you did. For example, these sentences need further explanation:

Report of the theoretical-practical assessment proposal from the perspective of EfS and IE for a school (school-based educational project, or SEP), prepared in the intermediate phase 

Classroom assessment design from the perspective of EfS and IE (CAD) 

It is not clear what you evaluated and how. 

These tasks have been explained in detail when explaining the intermediate phase (p. 6 and 7). The students' work represented both learning/assessment tasks and sources of information for research.

Similarly, you state: “After a first analysis of the data constructed, “but you did not specify what this analysis is based on.

The research team performed a first screening of the data on which the categories system was built.

This sentence is found in the results section, but would be better in the methodology section: dents evaluated the progression of their knowledge (QEPK) of assessment from the perspective of EfS and IE. The questionnaire consisted of 5 closed-ended questions in which the students had to rate different aspects of the knowledge worked on in the subject from 1 (not achieved) to 10 (fully achieved) and one open-ended question (Figure 2). A total of 24 students responded. 

We have moved it to the Methodology section.

Please explain how you grouped results in table 4 and table 5.

Table 4 only includes the percentages of the response options that were selected in question 3: “Rate your level of knowledge of the ethical, environmental, political and social dimension of assessment on a scale from 1 to 10”. None of the students selected response options from 1 to 5, and are therefore not included in Table 4.

Table 5 only includes the percentages of the response options selected in question 4: “Rate your level of knowledge of the inclusive dimension of assessment (community, attention to diversity, research, socio-environmental justice, participation) on a scale from 1 to 10”. None of the students selected response options from 1 to 6, and are therefore not included in Table 5.

As expressed above, some aspects of the methodology used need to be clearer explained in order to better interpret your results. Specially, the way you took information on the process during the intermediate phase is not clearly explained.  

This has been explained in the previous points

DISCUSSION

The discussion section has been improved by explaining what the weak and strong points consist of.

“This proposal is related to a context from which ideas can be extracted to be transferred to other similar scenarios or groups. Its principles and structure can serve as a reference that can be adapted to the different training contexts in which it can be put into practice. It will therefore be necessary for the teaching teams to integrate the principles of assessment from EfS and IE into their educational processes.”

4.1.

You state that “The progression of these ideas was tinged with superficial arguments related to the assessment processes, the verification of student learning, or the need for impartiality”. If it is a progression, such progression should be noted in your statement. This is, students ideas started with superficial ideas and along the process the complexity of argumentation was detected and so on.

We have improved this part by explaining the fact that ideas evolved to more complex levels, in contrast to the theoretical framework of reference.

 This sentence needs to be revised in terms of the way it is written: It is clear that EfS and IE have acted as guiding frameworks for the ideological and ethical dimension of the training of the students, future teachers 

As we mentioned earlier, a native English speaker specialised in correcting scientific papers has revised our paper and has changed the way this sentence was written.

Still, as I read it, it occurs to me that you could mention here projects as teachers for the future, and include as well as other references. It would also be important to mention the need for future research that you have identified.

We have included the need for future research.

 4.2

In sentence “As in the previous research objective, the ideas of the students in the initial phase were mainly found in non-complex levels” I suggest you change the word "found" to "categorised", as what you have done is a categorisation of the ideas of the students.

 We have changed “found” to “categorised”.

In sentence “This question is of great relevance in studies of this kind …”change “this question” by “this issue is of great relevance…”

 We have followed your recommendation.

Regarding paragraph: “In light of the above, it is considered that the perspectives of EfS and IE provided moral support, both in theory and in practice, and served not only to design assessment processes, but the subject as a whole. As expected, they were intentionally and consistently included in the designs of the assessment systems developed by the students, which is observed in their considerations.” First of all, I would like to point out that it serves as an example of the need for the article to distinguish between your course or teaching intervention and the topic you are proposing. This is a matter of linguistic editing, but it is important that you handle it to make your work clearer. Secondly, when you refer to "their considerations" you have to relate it to the assessment methods you have used in your research. That is, you have to refer to the evidence you have to draw conclusions. 

The way we understand educational processes, and more so in contexts of training future teachers, it is of paramount importance that the assumptions we defend and intend to develop in our students should also guide our teaching practice. This is why EfS and IE have been of support not only to the students, but also to the designs developed by the teachers.

With regard to the second comment, this sentence appears in the conclusions. The support regarding data collection instruments used in the research is stated in the results and conclusions sections. We have added an explanatory sentence in the conclusions to make it explicit.

CONCLUSIONS

I would suggest you to include a separate section with your general conclusions, limitations of the study and proposals for future research. 

It has been divided into two sections. On the one hand, the general conclusions, and, on the other hand, the limitations of the study and future research. The latter has been included in the conclusions (separate from the discussion section, as previously requested).

Here I suggest you to reinforce your current conclusions. In doing so, I propose you to avoid sentences as: “we can confirm” and to make statements more adjusted to what you actually have done: “Based on our results obtained… “ and here I proposes (but please adjust it to your real circumstances) “in this first didactic intervention focussed on the perspective of EfS and IE in the design and development of teaching assessments to solve the problems associated to Assessment…”

Thank you for your recommendation. We consider it has greatly improved the readability of the article. We have made the necessary changes in the General conclusions section.

On regard to the limitations of the study, I would propose to consider to mention that this has been a pilot implementation of your intervention, and that this analysis will serve you to validate some aspects of it and improve others. 

By including the need for future research, we have had to adapt the text. Thank you very much for pointing this out.

 Besides, in your current conclusions you say “In other words, …” but what you state is a new idea. So I suggest you restructure your conclusions reinforcing each of the two ideas you are defending. Furthermore, I think you should include in your conclusions and future research section the idea that this paper is “a contribution to help the education community to rethink not only their forms of evaluation, but also to understand the importance school evaluation systems may have in achieving sustainability and social justice” and point out further steps and research that are needed to continue working in that direction. 

As mentioned in the previous comment, adjustments have been made. Thank you very much for your comments to help us improve our research and dissemination.

REFERENCES

In the case of non-English references, which are abundant in this paper, please provide a translation into English so readers may understand the content of such reference. 

The guidelines of the journal have been followed. They do not specify the references have to be translated into English.

Apart from the link you include, please provide in all cases the name of the journal where it is published (e.g. In these references it is missing: 15, 16, 20, 21, 25, 28) an further information when needed (e.g. 17).

Thank you for your in-depth review. We have followed your recommendations.

Thank you, I am willing to read your improved version of the manuscript. 

Reviewer 4 Report

Dear authors

 As a scientific proposal, it would be desirable to better define the process to which students are entitled as students of an official degree, from what has been their research process, in which researchers can have a greater degree of freedom to guide the training induction. with an ideological bias.

All the orientations go in the direction of improving the guarantees of objectivity and ethical use of the teaching activity for research purposes.

Recommendations:

-It is recommended to adapt the abstract more explicitly to the IMDaR format. As well as reviewing the highly speculative and difficult title for the reader to find a contingency relationship with the analysis of an obligatory process of formative induction.

-The method is very confusing. There seems to be an overlap between research and the teaching process, and at no time is it made explicit how both processes are harmonized, and what were the guarantees for the students to be used in their right to receive education as an experience of not very explicit ideas. It is necessary to provide the explicit consent documents of the students to serve as a sample for these purposes, different from those that make them part of the student community.

The arguments from which the questionnaires arise, nor the concordant consistency between them, are not made explicit, leaving too much room for a priori speculation by the researchers.

-It is recommended to improve the contingency between the problem of the title and the categories of results.

 -It is recommended to differentiate between discussion and conclusions

Regards

Author Response

Thank you very much for your comments.

We greatly appreciate your efforts. We consider your contributions have helped to considerably improve the quality of our paper.

Below we present our responses to the different comments.

Recommendations:

-It is recommended to adapt the abstract more explicitly to the IMDaR format. As well as reviewing the highly speculative and difficult title for the reader to find a contingency relationship with the analysis of an obligatory process of formative induction.

We have made the necessary changes.

-The method is very confusing. There seems to be an overlap between research and the teaching process, and at no time is it made explicit how both processes are harmonized, and what were the guarantees for the students to be used in their right to receive education as an experience of not very explicit ideas. It is necessary to provide the explicit consent documents of the students to serve as a sample for these purposes, different from those that make them part of the student community.

This issue has been solved by adding an explanatory paragraph in the Methods section.

The arguments from which the questionnaires arise, nor the concordant consistency between them, are not made explicit, leaving too much room for a priori speculation by the researchers.

An explanatory paragraph in which these arguments are explained in section 2.2.

-It is recommended to improve the contingency between the problem of the title and the categories of results.

We have followed your recommendation.

-It is recommended to differentiate between discussion and conclusions

We have separated them in two different sections.

Regards

Round 2

Reviewer 3 Report

Congratulations, your paper quality has improved importantly since the previous version of the manuscript. Now it is clearer and appropiate for publication in sustainability journal.

Before publication, I suggest you make the following change: In the conclusions, you state "To conclude, based on the data analysed, the twofold purpose of teacher training regarding assessment from the perspective of EfS and IE has been achieved for a significant group of students.". However, you have not results on statistical significance, hence, I suggest you change the word significant for the percentage of student that achieved so.

Kind regards 

Author Response

Thank you again for your comments

Considering our analysis was more exhaustive than simply the percentages mentioned, we believe the appropriate statement in the conclusion, taking into account our research objective, should be the following:

To conclude, based on the data analysed, it can be said that the training process developed has had a positive impact on the students' ideas and knowledge regarding assessment from the perspective of EfS and IE.

Thank you very much for contributing to the quality of our paper.